# Electrospinning Polyvinyl Alcohol Reinforced with Chitin: The Effect of the Degree of Acetylation

**DOI:** 10.3390/polym16141955

**Published:** 2024-07-09

**Authors:** Andres Krumme, James D. Mendez

**Affiliations:** 1Department of Materials and Environmental Technology, Tallinn University of Technology, Ehitajate Tee 5, 19086 Tallinn, Estonia; andres.krumme@taltech.ee; 2Division of Science, Indiana University—Columbus, 4601 Central Ave., Columbus, IN 47203, USA

**Keywords:** chitin, chitosan, electrospinning, fungi, mushroom

## Abstract

Nanocomposites made via electrospinning were constructed of polyvinyl alcohol (PVA) and chitin. Chitin was extracted from a natural source (*Fomes fomentarius*), which allowed for precise control of the chemical properties of the resulting material. Chitin was chosen as a filler due to its low cost and widespread availability. Increasing the degree of acetylation of the chitin increased the Young’s Modulus of the resulting fiber mats but only at relatively high levels. While composites at lower acetylation levels were stable, no increase in the Young’s Modulus was observed, presumably due to decreased intermolecular bonding among fibers. The results suggest that precise control of the degree of acetylation of chitin, more than the loading amount and dispersibility, significantly impacts composite formation.

## 1. Introduction

Electrospinning is a technique that has evolved from a lab-based method for producing nano-scale fibers to a method capable of producing large quantities of fiber mats. The fundamental principle of electrospinning involves applying a high electrical potential to a polymer solution to draw out fibers with sizes ranging from nanometers to microns, which are then deposited on a substrate [1]. The substrates can vary depending on the specific use. Simple electrospinning setups deposit fibers on a flat surface. This method produces fiber mats with a random orientation of fibers. Aligning the fibers is possible by rotating the collection surface [2]. A rotating collector also has the benefit of using a greater effective surface area for collection. Electrospinning on specialized surfaces can produce more complex structures, such as coaxial fibers [3].

Basic electrospinning setups use a single syringe filled with the polymer solution, which is connected to a syringe pump and power supply. Although effective for producing fibers, these setups are very slow and essentially produce fiber mats in a batch approach [4]. Modern commercially viable systems use a wire as a charged electrode with a deposited polymer solution that provides multiple point sources (in the hundreds in some cases), increasing throughput [5]. Operating with a collection point set on constantly moving rollers, this system realizes a continuous production flow.

Electrospinning is a process that offers a significant advantage in producing materials with an exceptional surface area-to-volume ratio. This ratio increases as the size of the fibers produced through electrospinning decreases, leading to the creation of materials with even higher surface-area-to-volume ratios. This unique feature of electrospinning makes it a desirable technique for various applications where a high surface area is needed [6].

One application of electrospun fiber mats that has recently become more important is air filtration [7,8]. Key features of an effective air filter made from fibrous materials are small pore size, proper packing structure configuration, and small fiber size, all readily available with electrospinning [9]. Prior to 2020, studies focused on using electrospun fiber mats to filter the air in various settings outside of face coverings. One early application was in window screens [8]. Window screens need to be able to filter our unwanted containments while maintaining transparency. Liang et al. used a thermoplastic polyurethane (TPU) to produce a fiber mat to act as a transparent filter meant to be used as a window screen. They found that by controlling the concentration of the TPU in the spinning solution and, therefore, the fiber size, they could make a filter with almost 100% removal efficiency of micrometer-sized particles while losing very little transparency. In a similar fashion, Min et al. electrospun a composite of silk proteins with a comparable effect [10]. Not only were they able to make a transparent filter capable of filtering out micrometer-sized contaminants, but they could also control the fiber size to tune the transparency to only the visible spectrum. This had the benefit of blocking some infrared light, lowering the temperature of the interior space. 

Tebyetekerwa et al. were among the first to demonstrate the benefits of using electrospun fiber mats for face coverings [7]. Rather than innovating something completely different, they expanded on the basic concept already in use. Fabric- or cotton-based face coverings work because layers of fibers are combined to allow air to pass through while trapping harmful particles. Face masks made using electrospinning have the same configuration, but the fiber size can be tuned to maximize breathability and the ability to trap harmful particles. Face masks made with electrospun polyvinyl alcohol (PVA) have even been shown to absorb more moisture from the air and be more comfortable than traditional masks [11]. Recent studies have also shown that combining 3D printing and electrospinning of chitin can be used to create scaffolds with robust mechanical stability for this and a variety of uses [12].

In all of these applications, mechanical strength is imperative. Most of the examples discussed above use some backing material to reinforce the electrospun fiber mat, limiting the overall effectiveness. The silk protein filter described above was actually deposited on an existing wire mesh [10]. While this worked to create a stable product, it limited the transparency and increased the weight. Another approach to increasing mechanical strength is creating a composite with a polymer and a different stiff material. Carbon nanofibers are commonly used for this application [13]; however, they are expensive, challenging to process, and potentially hazardous [14]. Chitin nanofibers are a material with similar properties but are much more economical and readily available [15].

Chitin is the second most abundant biopolymer on Earth, found in anthropods and fungi. It is a linear polysaccharide with many hydroxyl groups that result in a high degree of intra- and intermolecular bonding, which means aggregates are common. In its natural state, it is used in a variety of food dishes and agricultural applications [16]. Extracted from various sources (mostly shrimp shells), it is used in a growing number of commercial applications. A key feature of chitin is its biocompatibility. Due to this fact, chitin has found uses in a variety of biomedical applications [17,18]. One such field is in wound healing and tissue engineering. Chitin-based materials have been shown to promote tissue growth and improve recovery times [19]. Increasingly, chitin is used in composites with traditional materials to compensate for some of the solubility issues, resulting in materials with improved mechanical properties [20]. 

Most of the applications described above use the fully deacetylated form of chitin, chitosan. This has the benefit of significantly increasing the solubility of the resulting material by decreasing the intermolecular bonds of the chitin. However, by reducing these interactions, the material’s mechanical properties also suffer a corresponding decrease. It has been shown that it is possible to precisely control the degree of acetylation (DA) of chitin by varying the reaction conditions during the extraction process. These resulting materials retain some of the mechanical properties of chitin and the increased solubility or dispersibility of chitosan [21]. 

This research aims to make a fiber mat composed of a nanocomposite of chitin and polyvinyl alcohol (PVA). To improve the mechanical properties, the degree of acetylation of the chitin/chitosan will be focused on.

## 2. Materials and Methods

### 2.1. Source Materials

The source of chitin used in this study was *Fomes fomentarius*, which was collected from forested areas around southwestern Tallinn, Estonia. This material is a tough bracket mushroom that grows over several years on birch trees and is chosen for its high chitin yield. The samples were rinsed with water to remove foreign material and dried for 24 h at 70 °C. A coarse powder of the material was made with a coffee grinder, and the samples were stored in sealed containers until needed. 

### 2.2. Extraction and Deacetylation

Chitin was extracted from the mushroom powder using a modified base–acid approach. Firstly, approximately 40 g of mushroom powder was added to 200 mL of a basic aqueous solution (sodium hydroxide) and refluxed to remove other organic material. To produce chitin with varying degrees of acetylation (DAs), the concentration of the base and reflux time were varied. Generally, a higher concentration of base and longer reaction time resulted in a lower DA. In this study, the concentration of sodium hydroxide varied from 2 M to 10 M, while the reaction time varied from one hour to eight hours (Table 1). This is slightly different from the conventional way of extracting the chitin first and then reacting it again with a base to produce chitosan. It was previously found that by adjusting the initial concentration of sodium hydroxide, the same control of the DA could be achieved without an additional step [21]. Many factors can affect the resulting DA, in addition to the concentration of sodium hydroxide and reaction time. For this reason, a specific DA was not targeted; instead, a range was used. Figure 1 shows the extremes of this reaction with pure chitin on the left and pure chitosan on the right. However, even under very mild extraction conditions, some level of deacetylation will occur. Alternatively, commercially available chitosan is rarely 100% deacetylated. 

After the initial base step was completed, the samples were neutralized by centrifuging them with deionized water approximately ten times. The wet samples were then mixed with 200 mL of a 2 M hydrochloric acid solution at room temperature to eliminate any inorganic components. The samples were once again centrifuged with DI water until they became neutral. Finally, the wet samples were treated with 200 mL of a 2% (by weight) sodium hypochlorite solution (bleach) for five minutes to remove any color from the sample. It is essential to wash the product quickly as a longer reaction time with bleach can degrade the chitin. 

Dry chitin is notoriously challenging to solubilize [22]. If allowed to dry, the samples produced here required multiple hours of mixing and sonicating in 1 M acetic acid to disperse. This process could be sped up by using a higher concentration of acid, but this added another step of reducing the acid concentration to the desired 1 M to remain consistent among samples. Instead, after centrifuging the sample back to neutral, it was stored in a concentrated solution of DI water (approximately 5–10% by weight). 

### 2.3. Degree of Acetylation Determination

A small film (radius of approximately 1 cm) was cast from the chitin solution to be tested via infrared spectroscopy on a Nicolet IR100 FT-IR or Interspec 200X. In both cases, attenuated total reflection (ATR) was used to obtain spectra of the chitin film. A carbonyl group is present in chitin (DA = 100%) but not chitosan (DA = 0%). Comparing the intensity of the absorbance peak at 1655 cm^−1^ from the carbonyl group to the hydroxyl peak at 3450 cm^−1^ was used to determine the degree of acetylation of the different chitin samples using the equation below [23,24]. Infrared spectroscopy was chosen to determine the degree of acetylation over other options based on the ability to return results quickly and the precision of the results.
$$DA\;\%=\frac{A_{1655}}{A_{3450}}\times \frac{1}{1.33}\times 100\%$$

At least five samples were tested for each batch of chitin, and the mean result was reported. The results from each sample were very similar, with an error of less than 5%, so no individual errors were reported in this study.

### 2.4. Fiber Morphology

Scanning electron microscopy (SEM) was performed with a JEOL 6390 LV (Japan Electron Optics Laboratory Company, Tokyo, Japan) at the Electron Microscopy Center at Indiana University–Purdue University Indianapolis to measure the individual electrospun fibers. The fiber mats were used directly with the SEM. It was found that it was easier to distinguish the fibers when imaging the edges of the fiber mats with fewer layers. However, fibers were also imaged at the center of the fiber mats, and no significant difference in size or morphology was noted. The instrument was run at 5 kV, with images obtained at magnifications from 3000 to 10,000. SEM was chosen over transmission electron microscopy (TEM) to demonstrate the morphology of the fiber mat better and visualize how the fibers stacked upon each other to form a fiber mat. TEM images could show the individual fibers in sufficient detail but made it more difficult to determine the precise layer of each fiber. Furthermore, as the fiber mats can be many layers deep, TEM was not able to be used for thicker samples. 

### 2.5. Electrospinning

Dry PVA and concentrated aqueous solutions of chitin were combined with 1 M acetic acid to produce solutions for electrospinning. As the chitin was already in a concentrated solution, the PVA was added to it to achieve the wanted concentration. Once combined, samples were stirred for at least one hour before being used. All percentages reported are weight percentages of the total solution. The concentration of acetic acid was kept constant at 1 M for solutions to improve the dispersion of chitin. Electrospinning was performed on an Inovenso Single Nozzle Electrospinning Machine (Inovenso Inc., Cambridge, MA, USA). The voltage and distance to the flat collector were kept constant at 14 kV and 10 cm, respectively. Electrospinning was allowed to continue at a rate of 0.25 mL/h until the fiber mat reached a thickness of at least 20 μm (approximately 8 h). 

### 2.6. Young’s Modulus Determination

Films were cut into small strips approximately 5 mm wide with a thickness of 0.06 mm. A single strip was clamped into an Instron Mechanical Tester (model # 2716-020). The maximum load was set at 100 N with a 40 mm/min speed. The Young’s Modulus for each sample was determined by taking the linear slope of the elastic region of the stress/strain curve. A minimum of 10 measurements were taken for each sample to obtain an average value. Figure 2 represents a stress/strain curve for a sample electrospun from a 1% chitin (DA of 41.8%) and 7% PVA in a 1 M acetic acid solution. The linear region where the Young’s Modulus was determined can be seen from 0% to approximately 2%. 

## 3. Results and Discussion

The effectiveness of chitin/chitosan as a composite material was investigated by combining chitin/chitosan with PVA. PVA was chosen as a carrier due to its practicality in electrospinning in an aqueous environment. Electrospinning aqueous solutions have several disadvantages over other solvents; primarily, the relatively high boiling point of water necessitates a prolonged rate of spinning to give adequate time for the solvent to evaporate before being deposited. Even given these constraints, PVA aqueous solutions have been demonstrated to work well with various electrospinning setups. A water-based solution is also necessary for the solubility of the chitin/chitosan, making PVA an ideal carrier polymer. 

Varying the ratio of chitin/chitosan to PVA and the DA of the chitin/chitosan demonstrated the effect of each property on the mechanical properties of the resulting fiber mat. The concentration of PVA was varied from 5% to 10%. This range was chosen based on the ability to spin the neat PVA at the high end and a combination with chitin/chitosan. Both 5% and 7% PVA would not form a stable mat without the addition of chitin/chitosan. Additionally, a concentration of over 10% PVA was not stable either. Below in Table 2 is a complete list of the average tensile strength of the fiber mat produced from each combination. When no data could be obtained, or a stable fiber mat was not achieved, a dash is used for the respective point on the graph.

To determine the role of the chitin/chitosan in the fiber matter, the amount of chitin/chitosan was varied along with the PVA concentration. In the first series of experiments, the amount of PVA in the solution was kept constant at 7%, while the amount of chitin was varied. These composites were formed with chitin/chitosan with DAs of 26.1% (Figure 3) and 35.5% (Figure 4). These two samples were chitosan and were able to disperse in the 1 M acetic acid solution at concentrations ranging from 0.5 to 2%. Composites with a higher DA were attempted but either could not disperse at higher concentrations (41.8% DA) or at any concentration (56.3% DA) to make a stable fiber mat. The 56.3% DA chitin and PVA solution appeared to disperse initially after stirring for one hour, but during the 8 h required to electrospin, a clear separation could be observed in the syringe. The same phenomenon was observed with 41.8% DA chitin above a concentration of 1%.

The expected and typical result would be that the Young’s Modulus of the resulting fiber mats would increase as the amount of the rigid filler (in this case, chitin/chitosan) was increased. Indeed, this has been observed many times before, even with a similar system of chitin and PVA by Junkasem et al. [25]. However, this was only observed for chitin with a higher DA. Solutions of PVA at 7% without chitin did not form fibers and are therefore excluded from this data set.

The lack of an increase in Young’s Modulus implies that the chitin with the lower DA does not act as a rigid filler. It does change the viscosity, but the lower DA increases the dispersibility and decreases the number of inter-fiber interactions. There is also a large degree of variation among the Young’s Modulus of the individual fiber mats for chitin with a DA of 26.1%. With only a slight increase in the chitin to a DA of 35.5%, the Young’s Modulus increases with the increase in chitin concentration consistent with a rigid filler. 

While the samples appear similar on a macro scale, the morphology of the fibers in the fiber mats is significantly different. Comparing the fiber mat with 26.1% chitin at 0.5% and 2% total loading, they both have similar-sized fibers and a large amount of beading (Figure 5). However, the fiber mat with 35.5% chitin changes drastically from 0.5% to 2% loading. Beading is evident at 0.5% chitin but is mostly gone when the total chitin content reaches 2%. Other studies have shown that viscosity has an outsized effect on fiber morphology (Figure 6) [26]. While the concentrations of chitin were the same, the higher DA of 35.5% resulted in a higher viscosity [27], resulting in this different structure. Close examination of the beads shows they are uniform, indicating beading from the electrospinning process of the carrier polymer, PVA. Insoluble chitin particles would not have a regular shape and would also not always appear on the fibers, which discounts this possibility. 

To determine if the PVA is playing a significant role in the resulting fiber mat, the amount of PVA was varied while holding the amount of chitin constant at 1% for both chitin with a 35.5% DA (Figure 7) and 26.1% DA (Figure 8). Using chitin with a DA of 35.5%, Young’s Modulus increased for the composites with 7% and 10% PVA. This result can be attributed to the low stability of the 5% PVA, as neat 5% PVA does not form a stable mat. Adding 1% chitin increases the stability enough to form a somewhat stable mat but is very near the threshold of stability, giving fiber mats that have a wide range of values. At higher concentrations of PVA, the mat is inherently stable, and the addition of chitin acts as a typical rigid filler. No significant change is observed between 7% and 10% PVA, presumably due to the fact that the 1% chitin can form a percolating network in each, and the additional PVA does nothing to change this. 

The chitin with a lower DA of 26% shows the opposite trend. The 5% PVA has the highest Young’s Modulus, with a sharp decrease for 7% and 10%. At this lower DA, the chitin is acting to increase the viscosity of the sample but also does not function fully as a rigid filler due to its increase in dispersibility. This is consistent with the results where the variation in this chitin has little to no effect on the resulting fiber mat. At lower concentrations of PVA, the increased viscosity allows for a stable mat to form, but at higher PVA concentrations, this added viscosity is more of a hindrance with little benefit as a rigid filler. 

To directly measure the effect of DA, fiber mat composites were electrospun at a PVA concentration of 10% and chitin concentration of 1%. We chose 10% PVA because the neat material showed the most stability when producing fiber mats, and 1% chitin was chosen because higher concentrations of chitin with a higher DA were not stable above 1%. The DA of the chitin/chitosan varied from 13.1% to 41.8%. Chitin with higher DAs was made but could not be solubilized in the mildly acidic solution used in this study. Young’s Modulus was mainly consistent at lower DAs before increasing at higher DAs (Figure 9). 

Similar to the previous results, the greater the DA, the higher the Young’s Modulus was overserved. As the higher DA corresponds with a more rigid filler, this result is consistent with studies demonstrating the link between the rigidity of a filler and the Young’s Modulus of the composite [28]. At lower concentrations of the DA, the chitin no longer functioned as a rigid filler, even though it increased the solution’s viscosity. Even as the Young’s Modulus increased, the variation among individual samples increased as well. This could be caused by increased aggregation among chitin fibers with a higher DA. In this case, individual samples or even different areas of a fiber mat could have local variations in the chitin content, causing more irregularity. 

## 4. Conclusions

The results show that the amount of chitin and the degree of acetylation can significantly impact the resulting electrospun fiber mats. While the Young’s Modulus increased with increased chitin/chitosan concentration with a DA of 35.5%, this did not occur with a DA of 26.1%. This result implies that the lower DA with decreased internal intermolecular bonding was not rigid enough to make a significant difference when incorporated with PVA. 

This relationship between the DA and Young’s Modulus of the fiber mat is most visible when comparing samples with the same chitin concentration but differing DA, as in Figure 9. A significant change in Young’s Modulus is seen once the highest DA is used. Essentially, this is a balancing act where a higher DA gives a higher Young’s Modulus but is too high, and the chitin will not disperse in a slightly acidic aqueous solution. It is expected that using chitin with a higher DA could increase the Young’s Modulus more but would also require the use of a stronger acidic solution. This would be difficult to achieve given the long time (8 h) required to produce a fiber mat.

The ideal composite found here has a chitin/chitosan content of 7–10% and a DA of approximately 35%. Higher concentrations of chitin could increase the Young’s Modulus further, but they are impossible to disperse without using stronger acids, which could degrade other portions of the composite. Similarly, a higher DA decreases dispersibility, making it impossible to achieve the ideal chitin concentrations.

## Figures and Tables

**Figure 1 polymers-16-01955-f001:**
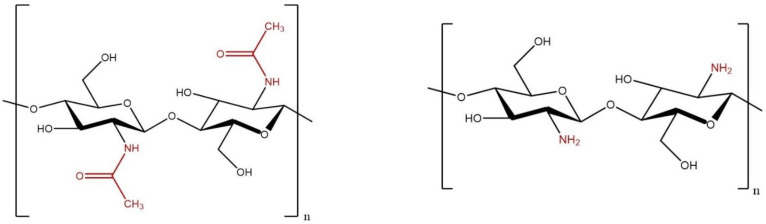
Idealized forms of chitin (**left**) and chitosan (**right**).

**Figure 2 polymers-16-01955-f002:**
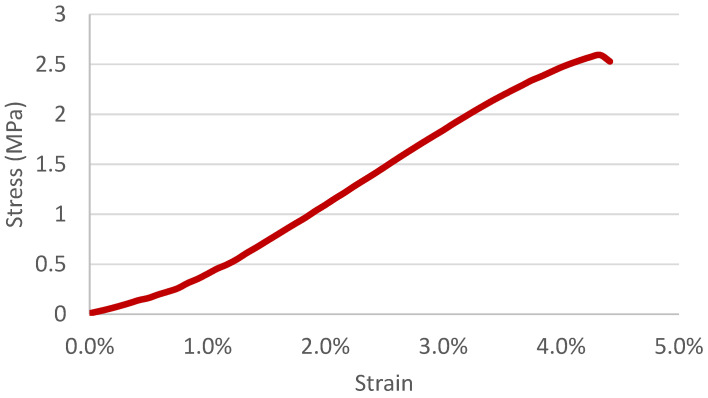
Characteristic stress/strain curve for chitin/chitosan and PVA electrospun films.

**Figure 3 polymers-16-01955-f003:**
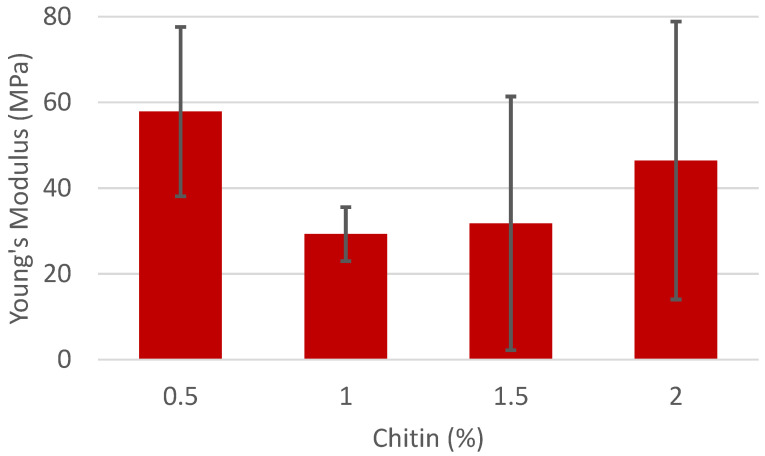
Young’s Modulus of fiber mats with 7% PVA and varying concentrations of chitin with a 26.1% degree of acetylation.

**Figure 4 polymers-16-01955-f004:**
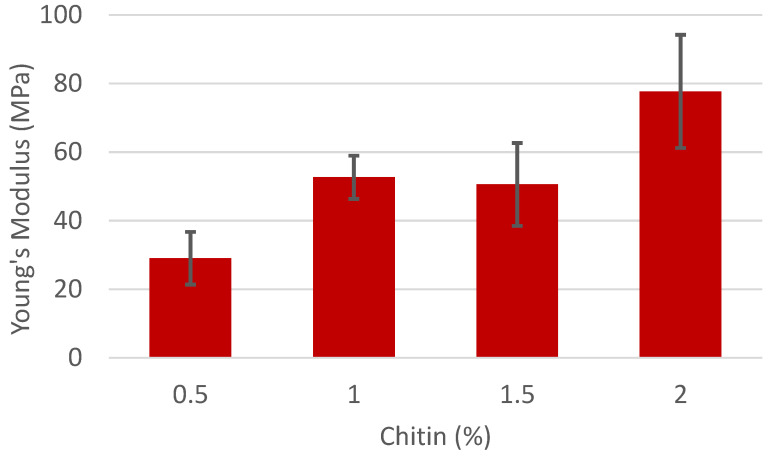
Young’s Modulus of fiber mats with 7% PVA and varying concentrations of chitin with a 35.5% degree of acetylation.

**Figure 5 polymers-16-01955-f005:**
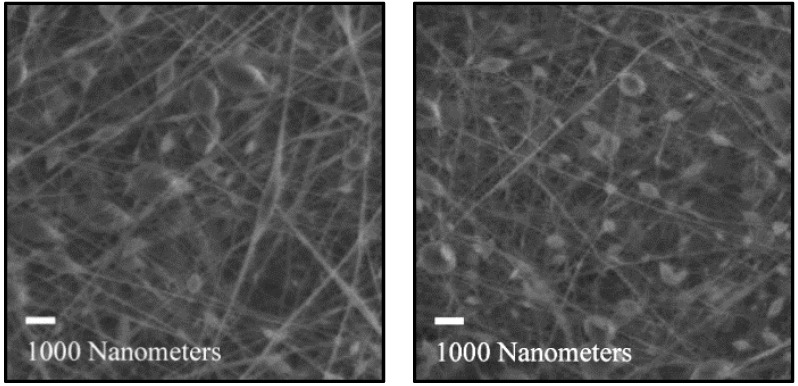
Electrospun fiber mats composed of chitin with a DA of 26.1% showed beading at both 0.5% (**left**) and 2% loading (**right**).

**Figure 6 polymers-16-01955-f006:**
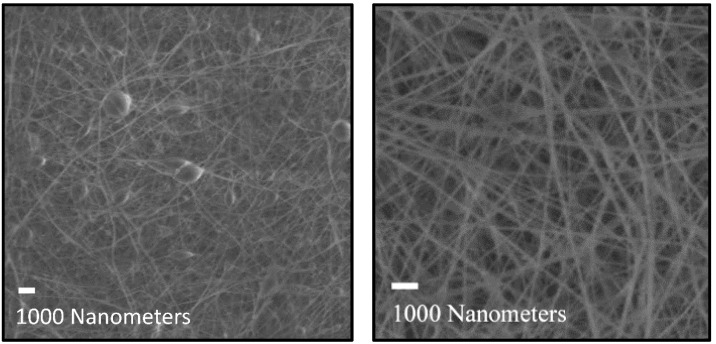
Electrospun fiber mats composed of chitin with a DA of 35.5% showed beading at 0.5% loading (**left**), similar to a lower DA, but smooth fibers were evident at 2% loading (**right**).

**Figure 7 polymers-16-01955-f007:**
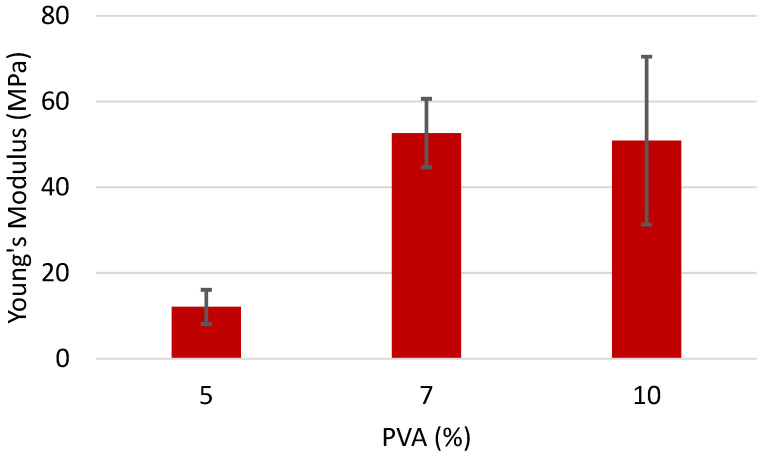
Young’s Modulus of fiber mats containing 1% of chitin (DA 35.5%) and varying concentrations of PVA.

**Figure 8 polymers-16-01955-f008:**
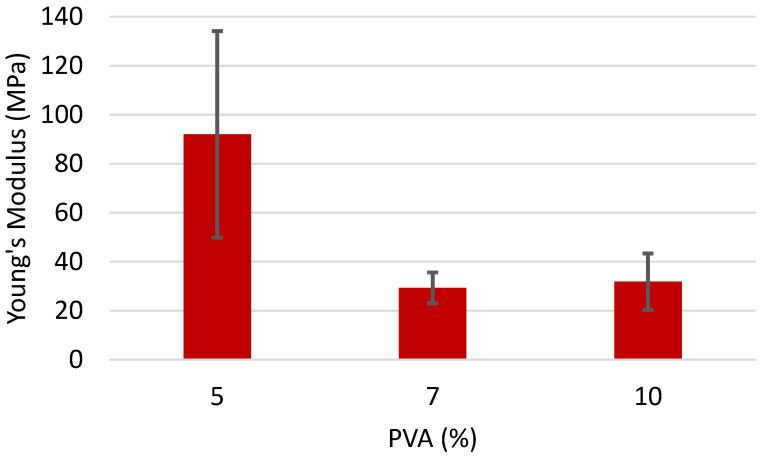
Young’s Modulus of fiber mats containing 1% of chitin (DA 26.1%) and varying concentrations of PVA.

**Figure 9 polymers-16-01955-f009:**
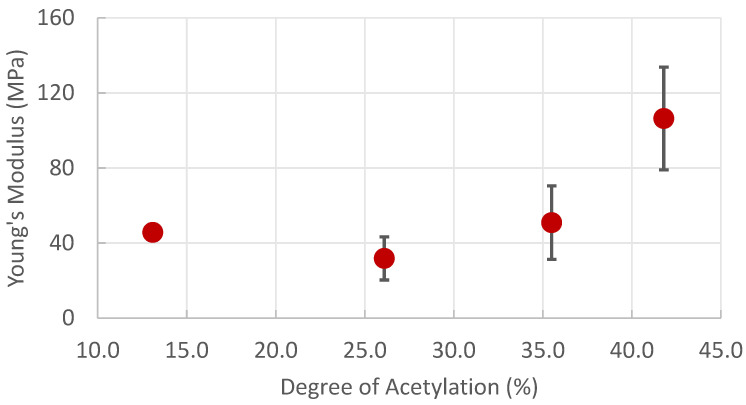
Young’s Modulus of fiber mats comprises 10% PVA and 1% chitin with varying degrees of acetylation.

**Table 1 polymers-16-01955-t001:** Variation in the DA based on base concentration (NaOH) and reaction time.

DA (%)	NaOH (M)	Time (h)
13.1%	10	8
26.1%	10	5
35.5%	5	3
41.8%	4	2

**Table 2 polymers-16-01955-t002:** Young’s Modulus data for electrospun fiber mats.

PVA (%)	DA (%)	Chitin (%)	Young’s Modulus (Pa)	Error (Pa)
5	-	-	-	-
7	-	-	-	-
10	-	-	1.74 × 10^7^	8.65 × 10^6^
5	41.8	1	-	-
7	41.8	1	-	-
10	41.8	1	1.06 × 10^8^	2.74 × 10^7^
5	26.1	1	9.20 × 10^7^	4.22 × 10^7^
7	26.1	1	2.93 × 10^7^	6.31 × 10^6^
10	26.1	1	3.18 × 10^7^	1.15 × 10^7^
5	35.5	1	1.21 × 10^7^	-
7	35.5	1	5.26 × 10^7^	3.16 × 10^6^
10	35.5	1	5.09 × 10^7^	1.96 × 10^7^
7	56.3	1	-	-
7	26.1	0.5	5.79 × 10^7^	1.97 × 10^7^
7	26.1	1.5	3.18 × 10^7^	2.96 × 10^7^
7	26.1	2	4.64 × 10^7^	3.24 × 10^7^
7	35.5	0.5	2.90 × 10^7^	7.68 × 10^6^
7	35.5	1.5	5.06 × 10^7^	1.21 × 10^7^
7	35.5	2	7.77 × 10^7^	1.65 × 10^7^

## Data Availability

The original contributions presented in the study are included in the article, further inquiries can be directed to the corresponding author.

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
