# Peer review of "Electrospinning Polyvinyl Alcohol Reinforced with Chitin: The Effect of the Degree of Acetylation"

_polymers, 2024, doi:10.3390/polym16141955_

Round 1
Reviewer 1 Report
Comments and Suggestions for Authors
Title: can be improved
Introduction: add more references in state of the art of this polymers reactions
Comments: add a scheme showing the reaction principale ingredients, especialy for the readers to understind more the reaction mecanisme.
Figures: not so clear
With kind regards
Author Response
Hello,
The following changes have been made to the manuscript based on the reviewer's suggestions.
Title - The title was changed to better reflect the actual experiment.
Introduction - A reference was added to describe the extraction process.
Other comments - The reaction scheme describing the deacetylation of chitin to chitosan was added to give the readers a reference.
Figures - The formatting of most of the pictures has been updated.
Reviewer 2 Report
Comments and Suggestions for Authors
Electrospinning has recently become increasingly popular as a powerful tool for creating new materials for biomedical applications. Among them, regenerative medicine (wound healing, tissue and organ cultivation, burn healing) is of particular importance. It is advisable to use biocompatible and biodegradable polymers for such materials, the synthesis of which is cheap, simple and amounts to millions of tons per year. Of course, there are few such polymers. But among them is a bright polymer, polyvinyl acetate. However, the mechanical characteristics of its materials obtained by electrospinning are quite modest, to put it mildly. Therefore, an important issue in polymer chemistry and materials science is to strengthen these materials and improve their mechanical properties. The authors solve this problem in a very original and elegant way, using the natural polymer chitin for this purpose. Chitin is a non-toxic, biocompatible and completely biodegradable polymer. Moreover, the authors use chitin from mushrooms. This is doubly interesting and significantly increases the novelty of the study, since there is much less work on mushroom chitin than on crab chitin. This work is important and interesting, it was carried out at a high scientific level. Statistics are present wherever required. The text of the work is bright, understandable, good English, and generally interesting to read. All conclusions seem correct and logical and do not contradict literary data and are confirmed by citations of relevant articles. The article is well illustrated, but figures 1,2,3,6-8 are simply terrible, it seems as if they were made by a schoolboy who is trying to do this for the first time. In addition, I kindly ask the authors to rewrite the abstract and add more quantitative data.
Author Response
Hello,
The manuscript has been updated to address the suggestions by the reviewer.
- Most of the figures have been reformatted to make the data presented easier to follow and understand.
- The abstract has been modified to make the reasons for the use of chitin more apparent.
- No additional quantitative data was available to add but hopefully the revised figures make the data that is available easier to read.
Reviewer 3 Report
Comments and Suggestions for Authors
The authors prepared nanocomposite fibrous membranes from polyvinyl alcohol (PVA) and chitosan by electrostatic spinning. In order to improve the mechanical properties of the fibrous membranes, the Young's modulus of the fibrous membranes was increased by controlling the degree of acetylation of chitin/chitosan. However, the work is not innovative enough and does not explain the specific reasons for the effect of the degree of acetylation on the mechanical properties, and expressions are unclear in many places, and after consideration, I recommend that this work be rejected. More specific reasons are given below:
1) The title of the article does not match the content, the title is Polyvinyl acetate, the content uses polyvinyl alcohol (PVA).
2) SEM graphs in the text are not clear enough.
3) ‘Two chitin samples with higher DA were produced, but either could not disperse at higher concentrations (41.8% DA) or at any concentration (56.3% DA) to make a stable fiber mat.’ The reference to ‘Two chitin samples’ is unclear.
4) ‘This result implies that the chitin with the lower DA does not act as a rigid filler.’ What does the conclusion refer to specifically? Lack of clear logic.
5) ‘Comparing the fiber mat with 26.1% chitin at 0.5% and 2%, they both have similar-sized fibers and a large amount of beading (Figure 4). However, the fiber mat with 35.5% chitin changes drastically from 0.5% to 2%. At 0.5%, a arge amount of beading is mainly gone at 2%. Other studies have shown that viscosity has an outsized effect on fiber morphology (Figure 5).’ Here 26.1% and 35.5% refer to DA, not chitin. Both concentration and degree of acetylation were variables in this experiment, but only the role of DA was explained in the explanation. There is also no in-depth explanation for the formation of beading.
6) ‘Using chitin with a DA of 35.5%, Young’s Modulus increased by 7% and 10% PVA.’ The expression here is not clear enough, does it refer specifically to the PVA content or Young's Modulus?
Comments on the Quality of English LanguageModerate editing of English language required.
Author Response
Hello,
The manuscript has been modified to address the reviewer's comments.
- Thank you for noticing this. I think it was a problem with autocorrect that has been fixed.
- While the SEM graphs could not be made more clear, discussions added for point 5 should make them easier for readers to interpret.
- This sentence was modified to make it easier to understand.
- This sentence was also modified to make the conclusion easier to understand.
- Additional information was added to the discussion of beading in the electrospun fibers.
- This sentence was missing a few words. It has been updated to make the meaning more clear.
Round 2
Reviewer 1 Report
Comments and Suggestions for Authors
The authors have performed all requierements
With regards
Author Response
The reviewer has stated that all of their comments were addressed.
Reviewer 2 Report
Comments and Suggestions for Authors
Figure 1 should definitely be corrected, since in the vast majority of cases chitosan contains both deacetylated and acetylated units, especially in this article. In its present form, figure 1 contradicts the text of the article.
Author Response
Hello,
The following change was made based on the reviewer's comment.
Figure 1 should definitely be corrected, since in the vast majority of cases chitosan contains both deacetylated and acetylated units, especially in this article. In its present form, figure 1 contradicts the text of the article.
Rather than present this as a reaction, both the structure of pure chitin and pure chitosan are presented. More context is given in the preceding paragraph to illustrate that these are idealized versions with the reality being a mix of both.
J.D. Mendez
Reviewer 3 Report
Comments and Suggestions for Authors
The authors prepared nanocomposite fibrous membranes from polyvinyl alcohol (PVA) and chitosan by electrostatic spinning. In order to improve the mechanical properties of the fibrous membranes, the Young's modulus of the fibrous membranes was increased by controlling the degree of acetylation of chitin/chitosan. Proposed publication after resolving some issues:
1) ‘Comparing the fiber mat with 26.1% chitin at 0.5% and 2%, they both have similar-sized fibers and a large amount of beading (Figure 5). However, the fiber mat with 35.5% chitin changes drastically from 0.5% to 2%. At 0.5%, a arge amount of beading is mainly gone at 2%. Other studies have shown that viscosity has an outsized effect on fiber morphology (Figure 6).’ Are 26.1% and 35.5% correct in describing chitin?
2) I still hope that the author can provide a clearer SEM picture of Figure 5 and Figure 6.
Comments on the Quality of English LanguageCheck the full text carefully for syntax problems.
Author Response
Hello,
The following changes have been made based on the reviewer's comments.
‘Comparing the fiber mat with 26.1% chitin at 0.5% and 2%, they both have similar-sized fibers and a large amount of beading (Figure 5). However, the fiber mat with 35.5% chitin changes drastically from 0.5% to 2%. At 0.5%, a arge amount of beading is mainly gone at 2%. Other studies have shown that viscosity has an outsized effect on fiber morphology (Figure 6).’ Are 26.1% and 35.5% correct in describing chitin?
This was unclear. The wording has been adjusted to make it clear that the 26.1% and 35.5% represent the degree of acetylation of the chitin used while the 0.5% and 2% represent the amount of chitin present in the composite.
I still hope that the author can provide a clearer SEM picture of Figure 5 and Figure 6.
I used the original file to improve the quality of Figure 6 (left) and adjusting various settings on the others.